# Traumatic Impact Assessment of CPR Load on a Human Ribcage

**DOI:** 10.3390/ijerph19063414

**Published:** 2022-03-14

**Authors:** Luis Antonio Aguilar-Pérez, Christopher René Torres-SanMiguel, Marco Ceccarelli, Guillermo Manuel Urriolagoitia-Calderón

**Affiliations:** 1Instituto Politécnico Nacional, Escuela Superior de Ingeniería Mecánica y Eléctrica, Sección de Estudios de Posgrado e Investigación, Unidad Zacatenco, Mexico City 07738, Mexico; laguilarp@ipn.mx (L.A.A.-P.); gurriolagoitiac@ipn.mx (G.M.U.-C.); 2Laboratory of Robot Mechatronics, Department of Industrial Engineering, University of Rome Tor Vergata, 00133 Rome, Italy; marco.ceccarelli@uniroma2.it

**Keywords:** biomechanics, FEM, rib stiffness, CT

## Abstract

Chest compression is a parameter of injury criteria assessment for human beings. Additionally, it is used to find the external compression response as a result of vehicle crashes, falls, or sporting impacts. This behavioral feature is described by many deterministic models related to specific experimental tests, hindering distinct scenarios. The present study evaluates the energy absorbed as a function of rib compression. The proposed model was obtained from three different computed tomography (CT) studies. The anthropometric values are interpolated to obtain a parametric curve for a human rib’s average size. The computed results are compared against an STL-DICOM^®^ file used to obtain a virtual reconstruction of one rib. A numerical model of the behavior of the thorax displacement expressed injury in the human rib model’s stiffness. The proposed model is used to determine the correlation of the input payload versus the numerical stiffness value. The outcome is confirmed by numerical analyses applied to a virtual human rib reconstruction.

## 1. Introduction

One of the essential parts of a human being is the thorax. The ribcage’s primary function is to protect the internal soft tissues from local injuries and over-compressions applied to this area. Thorax injury severity has been summarized as a comparative of biomechanical parameters, such as displacements against traumatic events reported by medical professionals [1]. Human chest compression represented by 33.3% of the length causes failure in the patient’s thorax, fractures on both sides, or even pneumothorax or hemothorax. The injury criterion [2] combines statistical data that correlates the probability of this kind of damage in the human body with the physical variables related to the traumatic event. Several criteria, such as those previously addressed in this paper, comparatively classify systems according to an abbreviated injury scale 3+ (AIS), ranging from medium to dangerous, if the frontal compression exceed specific values. For example, the compression criteria indicate that the total compression must always be below 52 mm [3]. The criteria report that, upon collision, the thorax can experience force values below 3.3 kN [3]. The thorax’s injuries can be reproduced in laboratory conditions through artificial surrogates, biological surrogates, or computational simulations. The development of artificial test devices substitutes biological elements in the laboratory for real conditions, even if some random parameters cannot be evaluated. For example, through Dead Human Surrogates (DHSs) it can estimate the physical responses for wide ranges of events, including parameters like cardiopulmonary pressure, the natural degradation of tissues, or even the absence of muscular tension that causes a variation in almost 20% of the results [4,5,6]. In the same way, anthropometric test devices (ATDs) can then reproduce those results to confirm the initial behavior of these devices. Thus, the natural geometry complexity of the ribs can be overestimated in relation to the real injuries sustained by the thorax. The Hybrid III model’s biomechanical response, described by [5], reports a value four times higher than the real one’s stiffness. Perz in [7] noted that, in almost 40% of cases, most of the models overestimate the inertia values of different components that replicate the thorax’s behavior. In this way, the device’s reliability must be tuned to the real mechanical properties of durability and repeatability. Bone stiffness can be described as the strength of the entire bone structure, but, on the other hand, material stiffness is defined as the ability to oppose a mechanical load, such as compression or bending. This can be changed by material properties, the element’s geometry, and the boundary conditions used during the analysis. Moreover, in material engineering, the property used to oppose fracture propagation is known as tenacity, but these parameters can be overestimated in brittle materials, such as the bone. The human thorax in relation to ATDs tries to reproduce the ribs’ mechanical properties, vertebrae, breastbone, and cartilages using common materials, such as steel, plastic, or foam, with simplified geometries and standard cross transversal sections.

This paper evaluates the human rib’s complex geometry by developing a standard model that considers the element’s material properties and inertia values. The comparative results are used as validation criteria for the developed model. Once this numerical model is confirmed, a correlation between the payload is applied for one rib versus the rib stiffness computed from the numerical model. Finally, the computed model develops a dummy training system for educational purposes.

## 2. Materials and Methods

The human rib’s mechanical stiffness is a design parameter for artificial human ribs used by artificial surrogates. The proposed method uses a second-grade geometric curve computed through a dataset. The second-grade geometric curve was used as a guideline in a CAD sweep operation. Table 1 summarizes the entire dataset used alongside this work. All the data used here were obtained from the corresponding test subjects through letters of informed consent and anonymized through our research institution’s internal procedure.

Reference [8] reported the procedure to obtain the guideline curvature of the rib. Figure 1 compares both models to describe the reconstructed guideline curvature of the rib.

The main improvement of this model is that the entire guideline curvature is interpolated by a parametric curve. In addition, the proposed model includes a rib variation of its cross-section extensively described in [9] and mathematically explained in [10]. The second-grade curve is defined by the curvature of the rib as follows:(1)$$Ax^{2}+2Bxy+Cy^{2}+2Dx+2Fy+G=0$$
where the letters “*A*, *B*, *C*, *D*, *F*” and “**G**” are the constants computed by
$$E=\left[B\;C\;D\;F\;G\right]=\frac{M}{x^{2}\;}$$
$$M=\left[\begin{array}{ccc} 2xy & y2 & \begin{array}{ccc} 2x & 2y & 1 \end{array} \end{array}\right]$$

These parameters are substituted into the general ellipse form as follows:(2)$${\left(\frac{x-h}{2*B}\right)}^{2}+{\left(\frac{y-k}{2*A}\right)}^{2}=0,$$
where
$$h=\frac{\left(C*D-B*F\right)}{B^{2}-A*C};k=\frac{\left(A*F-B*D\right)}{B^{2}-A*C}$$

Then, the radius of the curvature is solved by:(3)r=B2(B2A2−1)cos2(θ)+1 ,

On the other hand, the cross-section of the rib is described by the following:(4)$$T_{sect}=\left[\begin{array}{c} d_{men}*\sin\left(\theta \right)\\ D_{may}*\cos\left(\theta \right)\\ 0 \end{array}\right],$$

By using the previous equations, the deformation of the rib is computed as follows:(5)$$U_{T}=U_{N}+U_{S}+U_{M}+U_{MN},$$
where
$$U_{N}=\int_{0}^{\frac{7\pi }{6}}\frac{B*F_{Q}^{2}*\sin^{2}\left(\theta \right)}{2*A_{area}*E_{bone}*r}\;d\theta$$
$$U_{S}=\int_{0}^{\frac{7\pi }{6}}\frac{B*F_{Q}^{2}*k*\cos^{2}\left(\theta \right)}{2*A_{area}*G_{bone}*r}\;d\theta$$
$$U_{M}=\int_{0}^{\frac{7\pi }{6}}\frac{B^{2}*F_{Q}^{2}{\left(\cos\left(\theta \right)-1\right)}^{2}}{2*I_{xx}*E_{bone}\left(\left(\frac{B^{2}}{A^{2}}-1\right)*\cos^{2}\left(\theta \right)+1\right)}d\theta$$
$$U_{MN}=\int_{0}^{\frac{7\pi }{6}}\frac{-B*F_{Q}*\sin\left(\theta \right)\left(\cos\left(\theta \right)-1\right)}{A_{area}*E_{bone}*r}d\theta$$

The rib stiffness is found as reference [11], where the linear spring constant relation can approximate the load deflection of a body, *k_Stiffness_*. Then, the magnitude of the energy that a body can absorb is a function of its elongation, “*δ*”. Additionally, in reference [12], the main effects of the rib’s material properties are mentioned under injury conditions. For example, when testing a rib with no fractures, it was observed that the yield stress was around 69.64 MPa, but when a rib had a fracture, the yield stress value was 59.33 MPa. Table 2 shows several authors’ mechanical properties as the minimum values required to fracture a rib in static conditions.

Castigliano’s second theorem is computed to obtain the deformation caused by the payload Fq shown in Figure 1, which is given by:(6)$$\delta U_{T}=\delta U_{N}+\delta U_{S}+\delta U_{M},$$
where
$$\delta U_{N}=\int_{0}^{\frac{7\pi }{6}}\frac{B*F_{Q}\sin^{2}\left(\theta \right)}{A_{area}*E_{bone}*r}d\theta$$
$$\delta U_{s}=\int_{0}^{\frac{7\pi }{6}}\frac{B*F_{Q}*k*\cos^{2}\left(\theta \right)}{A_{area}*G_{bone}*r}d\theta$$
$$\delta U_{M}=\int_{0}^{\frac{7\pi }{6}}\frac{B^{3}F_{Q}{\left(\cos\left(\theta \right)-1\right)}^{2}}{I_{xx}E_{bone}{\left(\left(\frac{B^{2}}{A^{2}}-1\right)*\cos^{2}\left(\theta \right)+1\right)}^{\frac{3}{2}}}d\theta$$

Finally, the mechanical stiffness of a human rib can be computed by Equation (7):(7)$$k_{stiffness}=\frac{2*U_{T}}{\delta U_{T}^{2}},$$

In addition, the circumferential stress is computed by:$$\sigma_{\theta \theta }=\frac{F_{Q}\sin\theta }{A_{area}}+\frac{B*F_{Q}\left(\cos\theta -1\right)\left(A_{area}-\frac{2*\pi *D_{M}*y*\alpha }{D_{m}}\right)}{A_{area}*y*\left(A_{area}-\frac{2*\pi *D_{M}*y*\alpha_{1}}{D_{m}}\right)*\beta }\;\;\alpha =\frac{B}{\beta }-\sqrt{\frac{B}{\left(\frac{B^{2}}{A^{2}}-1\right)\cos^{2}\theta +1}-D_{m}^{2}}\;\;\beta =\sqrt{\left(\frac{B^{2}}{A^{2}}-1\right)\cos^{2}\theta +1}$$

## 3. Numerical Procedure

The computed geometrical reconstruction’s numerical values are then compared; an FEM model analysis was performed using the data in Table 1 and substituted into Equations (1)–(3). The numerical simulation was run on MATLAB^®^ using an open-source toolbox [23] to compile the analysis. Figure 2 shows the rib’s computed guideline by using Equations (1) and (2).

The computed parametric curve is a guided line on a sweep CAD operation with a controlled tetrahedral mesh. The image used to parametrize the cross-section of the rib reports a voxel size of 0.97 mm. This size could decrease the model’s fidelity by a maximum error of 7%. In this way, an elliptical section was used to approximate the variable shape of the rib. The cross-section of the rib can be described as a ratio of the cortical and cancellous tissues. In [24], this ratio varies from 5% at the tip to 75% in the costovertebral joint. As is mentioned in [12], the rib cross-section was simplified to a solid elliptical shape, and [25] represents a cancellous ratio for the cortical bone of 0.5. In addition, by using Table 2, the model’s material properties can be considered as orthotropic, with a Young’s modulus value of 11.04 GPa, which corresponds to the average value reported. Figure 3 shows a magnified view of the obtained results. First, the ends of the reconstructed sweep CAD operation were computed by using Equation (4). Next, the elements meshed in size and orientations were calculated using the open-source software [26]. The mesh result was obtained under 0.25 s on a laptop with Core i5 8th generation and 16 Gb of ram. The total elements were 1652 tetrahedra and 3802 faces. In this way, we can develop a master curve for the numerical model that can be validated against the tomography model in terms of the solving times and size of the model.

In addition, by using the MATLAB^®^ code, a DICOM^®^ file from the initial dataset was used to build the seventh rib. The right seventh rib has a surface meshed with no control of each element’s orientation, position, or size. Therefore, the STL file has to be reduced from 12.5 MB to 1.51 MB. This reduction was applied because the MATLAB^®^ files for this version v7 were only allowed to work with files with a maximum size of 2 MB. Finally, the tetrahedral mesh was carried out, obtaining a total time of 2.846 s. The total elements obtained were 173,929 tetrahedra and 359,634 faces. The model obtained is shown in Figure 4.

Thus, the simplified reconstructed model was named Model A. This model is shown in Figure 5a. Similarly, Model B represents a virtual model of the reconstructed STL file from the DICOM^®^ tomographic files. This model is shown in Figure 5b. Finally, the results obtained using Equations (5)–(7) are named Model C. The boundary conditions were similar for both geometrical models; the left end of the rib is fixed in all directions, and only a compression force is applied in the “X” direction. On Model A, the left end of the sweep CAD model is fixed in all directions. Therefore, the payload Fq value is applied to the nodes at the right end on Model A. The costovertebral joint is fixed in all directions on Model B. For Model B, the payload Fq value is applied on the nodes near the cartilage section only in the “X” direction. The payload Fq was 180 N; the reported values’ average results are shown in Table 3.

Model A’s computed elapsed time was 0.3 s against Model B, which was 29 min and 28 s. The maximum displacement obtained is shown in Figure 6. First, the nodal distance for Model A ($D_{d_{A}}$ = 49.20 mm @ 37.57° node ($P_{A_{i}}\left(206,55\right)$ and node $P_{A_{f}}\left(167,25\right)$)) was measured, and the value was compared against the computed results obtained for Model B ($D_{d_{B}}$ = 53.82 mm @ 54.83° node ($P_{B_{i}}\left(200,-13\right)$ and node $P_{B_{f}}\left(169,-57\right)$)). The error was around 8.6%. Then, by comparing the same results by using Equation (6) of Model C, the computed results were ($D_{dC}$ = 48.24 mm $P_{C_{i}}$(192.8, 56.38),$\;P_{C_{f}}$(166.4, 16.0)). This point has an error of 12.01%. The numerical values shown in Figure 6a represent the reaction points computed through Equation (6). Figure 6b represents the computed values for Model A. Figure 6c represents Model B’s computed values.

Equation (7) of the proposed model estimates the stress–strain that remains upon yielding the bone’s stress value. However, this consideration only applies upon the assumption that the material is orthotropic and lineal, as is shown in Figure 7. It is then verifiable if the existing areas in the STL file model, or the reconstructed rib model, reach the yield stress value reported in Table 2.

The model constructed by the geometrical rib sweep does not present stress values near the yield stress reported in Table 2. In this way, the proposed model is used with certainty to evaluate parameters, such as material conditions (i.e., what if the rib geometry was reconstructed by steel or 3D printed PLA plastic for dummy purposes), the force applied (i.e., how the payload applied affects the displacement of the rib), and, finally, how the modification of the rib inertia values can improve the stiffness of artificial ribs. Thus, 32 simulations were performed in order to evaluate such conditions. The computed results are summarized in Table 3. In addition, we grouped the computed results into six sets of conditions. The first and second sets represent the model’s condition constructed by a geometrical sweep with the main axis of the ellipse in the horizontal and vertical positions, both considering 3D printed PLA plastic. The third and fourth sets of values represent the condition of the model constructed by a geometrical sweep, with the main axis of the ellipse in the horizontal and vertical positions, but this time considering the steel material properties. Finally, the fifth and sixth sets of values represent the computed results of the model constructed by the proposed geometrical sweep, but this time by considering a scaling of two times the original dimensions of the initially measured cross-section of the rib. These considerations were made to evaluate the best way to manufacture the element and also reproduce the biomechanical response.

Figure 8a shows the displacement graphic versus the applied payload. Figure 8b shows the strain–stress curve of the six sets of values. From Figure 8b, we can observe that the model is constructed by a geometrical sweep considering the main axis of the ellipse parallel to the main vertical axis, and the response is closely similar to the biomechanical behavior of the bone. However, in Figure 8a, we can observe that such a response is not similar to the one obtained from the bone material. In contrast, the computed results for the same model use the material properties of 3D printed plastic and consider an increase in the dimensions of the cross-section and the horizontal axis, which is parallel to the main vertical axis and has a similar behavior to the biomechanical response of the payload displacement behavior.

## 4. Discussion

The model is used to correlate the energy that a rib’s geometry can absorb as a function of its compression, so that it correlates with the material properties of the element and the inertia values related to their cross-section. Even if the proposed model is restricted to the used dataset, the numerical results obtained in this work were consistent in different scenarios and software methods. The model only evaluates the rib’s cross-section as a solid elliptical shape, changing this geometry to modify the porosity’s internal value or the cross-section shape, such as commercial dummies used to compare the results. The model modifies these parameters by changing Equations (2) and (4). The chosen values for the compression limits are described in [3] as the reference parameters to medical conditions, such as fractured ribs, the hemithorax, or pneumothorax. Using the reported values for such criteria was to permanently remain under yielding stress bone material properties to develop a linear stiffness behavior, such as a compressional spring. The rate of compression can be classified as medium-to-light damage. Such a conclusion is as a result of the maximum displacement criteria pointing out a displacement of 52 mm, prior to internal damage or bone fracture [3], and these values can be attributed to the non-breastbone interconnection to other ribs or dynamical conditions. Equations (5) and (6) will be modified to include this interconnection for the entire ribcage in future works. The system’s parameters are modified by changing some anthropometric values to significantly influence the ribcage stiffness. One parameter not analyzed in this paper is vibration, a dangerous factor for the human body. Using the results, it is possible to define the essential anthropometric values that were considered to determine these vibration modes on different systems.

## 5. Conclusions

This mathematical model uses a dataset to average the geometrical conditions to reconstruct a sweep CAD model and then evaluate the rib’s response under compression caused by a payload of 180 N. The response is characterized as a function of rib stiffness, which is determined by the characteristics of the bone material, rib shape, and the payload value. The numerical results obtained with the proposed formulation establish a straightforward way to understand how the anthropometric values modify ribcage stiffness. These findings suggest an essential role of this parameter on the material used to manufacture artificial dummies. This research has some limitations. The main limitation is the tomographic study. The model’s voxel size does not allow for the evaluation of the cancellous–spongy tissue behavior. In future works, this interaction will be studied. However, given the concern for the small sample, this work highlighted that some dummies’ material properties could represent a serious modifier of real thorax behavior. One way to control this parameter is by manipulating the cross-section, which can increase the ribcage stiffness by almost double, by modifying its shape. Future work will use the mathematical development presented in this paper to evaluate the ribcage stiffness by assuming different material properties and element conditions as a modifier of their response and obtaining a closer biomechanical response to be reproduced in artificial dummies.

## Figures and Tables

**Figure 1 ijerph-19-03414-f001:**
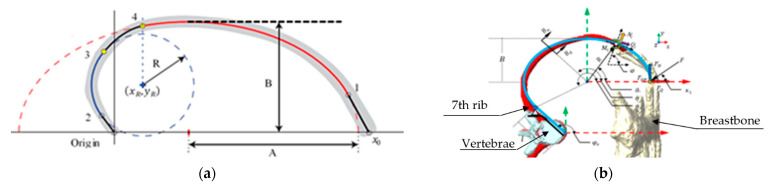
Parametric curve of the rib: (**a**) horizontal plane view; (**b**) sagittal plane view.

**Figure 2 ijerph-19-03414-f002:**
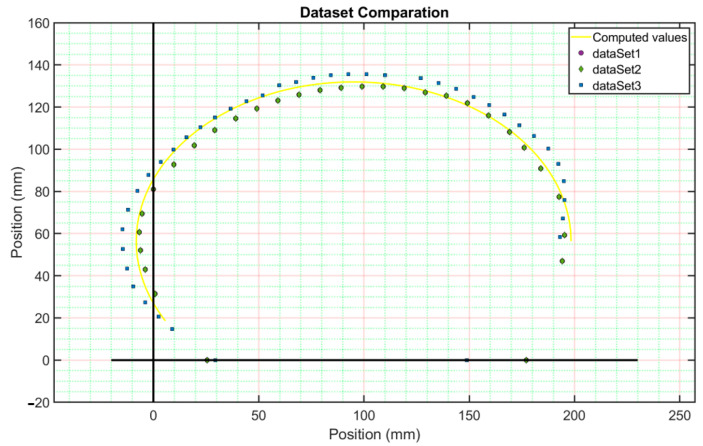
Comparative computed guided line values of the rib’s reconstruction against the dataset used.

**Figure 3 ijerph-19-03414-f003:**
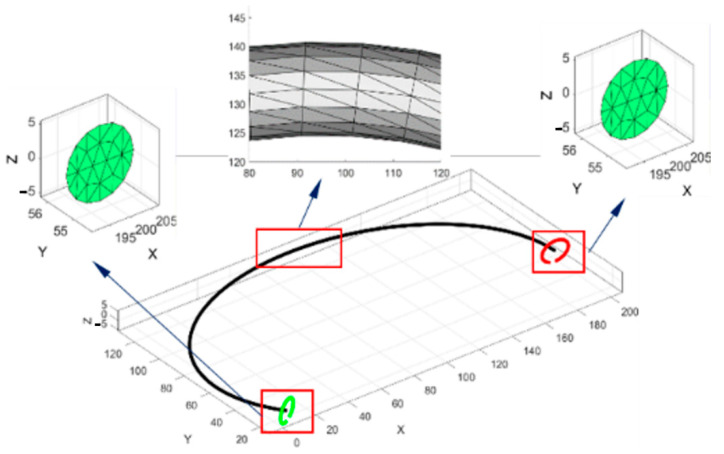
Local controlled mesh computed with a guided line of the rib reconstruction.

**Figure 4 ijerph-19-03414-f004:**
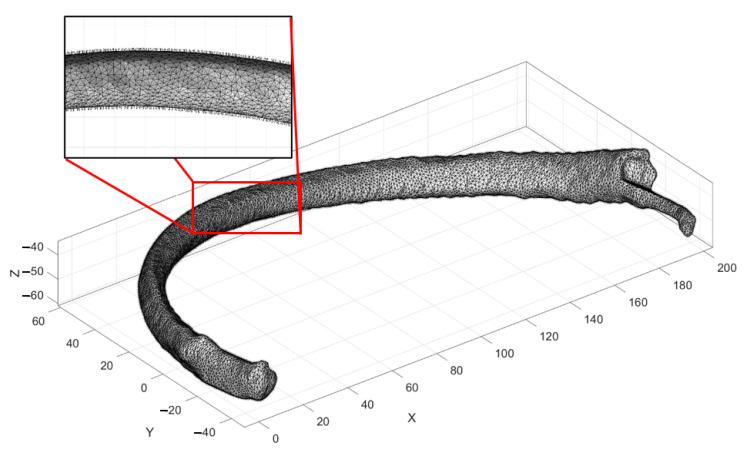
The model of the biomechanical tomographic images.

**Figure 5 ijerph-19-03414-f005:**
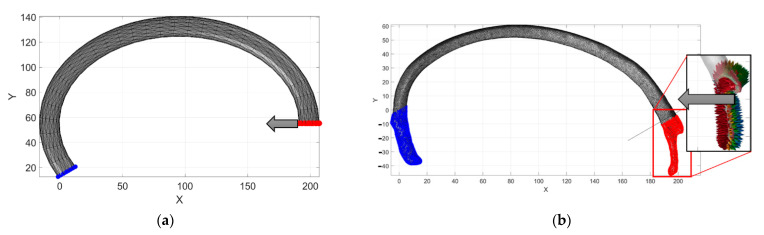
Biomechanical boundary conditions: (**a**) proposed reconstructed model; (**b**) STL rib model.

**Figure 6 ijerph-19-03414-f006:**
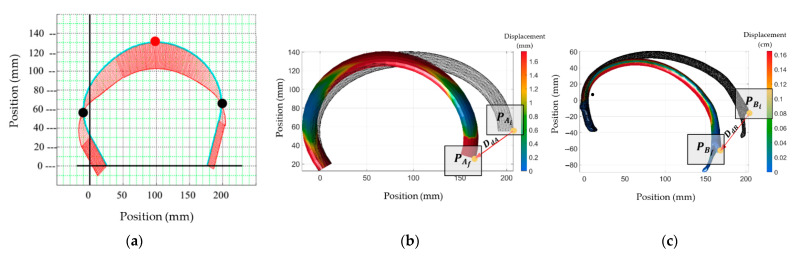
The biomechanical model’s boundary conditions: (**a**) own developed model; (**b**) geometrical rib reconstruction; (**c**) STL model of the rib.

**Figure 7 ijerph-19-03414-f007:**
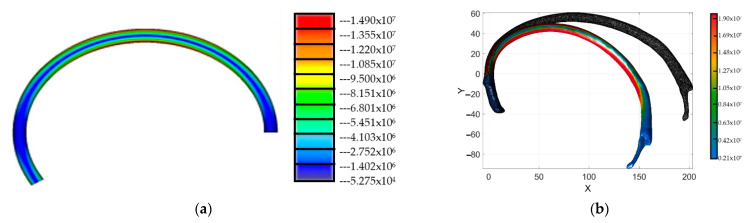
The biomechanical model’s stress distribution: (**a**) geometrical rib reconstruction; (**b**) STL model of the rib.

**Figure 8 ijerph-19-03414-f008:**
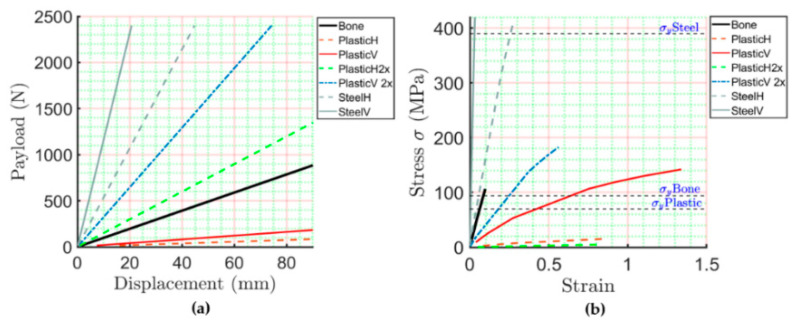
Comparative analysis of the computed results. (**a**) displacement versus payload; (**b**) strain versus stress.

**Table 1 ijerph-19-03414-t001:** Morphological parameters from 4 tomographic studies.

	Dataset 1	Dataset 2	Dataset 3	Dataset 4
Age	75	21	43	22
Genre	M	M	F	M
**Voxel size**				
Z (mm)	1.5	2	4	1
X (mm)	0.68	0.97	0.68	0.97
Y (mm)	0.68	0.97	0.68	0.97
Wide side (W, mm)	96.04	106	93.5	99
Short side (H, mm)	63.00	73.5	74.5	73
**The Radius of the Cross-Section**				
Z1 (mm)	16.22	20.70	19	17.7
Z2 (mm)	11.02	12.80	7.34	8.56
Y1 (mm)	10.48	11.50	10.5	11.6
Y2 (mm)	4.88	4.40	2.51	3.79

**Table 2 ijerph-19-03414-t002:** Values reported for orthotropic material bone properties.

Author	Maximum Payload Application Reported before Breaking	Young’s Module (E, GPa)	Velocity (mm/min)	σu (MPa)
Yoganandan [13]	153	2.37	2.50	2.102
Roth [14]	---	14.00	---	70.00
Currey [15]	---	13.00	---	110.00
Kieser [16]	150	4.70	10.00	53.33
Pezowics [17]	---	5.97	---	---
Gilchrist [18]	---	11.50	---	---
Forbes [19]	---	26.00	---	---
Stein y Granik [20]	226.80	11.50	2.54	106.00
Martínez-Sáez [21]	---	7.50	1.70	---
Goumtcha [22]	---	14.00	---	70.00

**Table 3 ijerph-19-03414-t003:** Computed values for the model constructed by a geometrical rib sweep.

		Payload (N)	δx (mm)	Ut (J)	k
The main vertical axis of the rib cross-section	BONE	15	1.5266	0.2149	0.1844
		30	3.0532	0.8597	
		45	4.5797	1.9343	
		90	9.1595	7.7372	
		180	18.319	30.949	
		200	20.3544	38.2086	
		220	22.3899	46.2324	
		240	24.4253	55.0204	
	PLASTIC	15	7.3866	0.8321	0.0305
		30	14.7732	3.3284	
		45	22.1598	7.4889	
		90	44.3195	29.9554	
		180	88.6391	119.8217	
		200	98.4879	147.928	
		220	108.3366	178.9929	
		240	118.1854	213.0164	
	STEEL	15	0.1294	0.0157	1.8734
		30	0.2588	0.0628	
		45	0.3882	0.1412	
		90	0.7765	0.5648	
		180	1.553	2.2591	
		200	1.7255	2.789	
		220	1.8981	3.3747	
		240	2.0706	4.0162	
	PLASTIC 2 × the cross-section	15	0.4641	0.0846	0.786
		30	0.9282	0.3386	
		45	1.3923	0.7618	
		90	2.7847	3.0473	
		180	5.5693	12.1892	
		200	6.1882	15.0484	
		220	6.807	18.2086	
		240	7.4258	21.6697	
Horizontal main axis of the cross rib section	PLASTIC	15	1.5917	15.9855	0.0125
		30	6.3667	31.9709	
		45	14.325	47.9564	
		90	57.3	95.9128	
		180	229.1999	191.8256	
		200	282.9628	213.1396	
		220	342.385	234.4535	
		240	407.4664	255.7675	
	STEEL	15	0.029	0.2799	0.7399
		30	0.1159	0.5598	
		45	0.2608	0.8397	
		90	1.0433	1.6794	
		180	4.1732	3.3588	
		200	5.1521	3.7319	
		220	6.2341	4.1051	
		240	7.419	4.4783	
	PLASTIC 2 × the cross-section	15	0.1321	1.0015	0.2634
		30	0.5285	2.0031	
		45	1.1891	3.0046	
		90	4.7563	6.0093	
		180	19.0254	12.0185	
		200	23.4881	13.3539	
		220	28.4206	14.6893	
		240	33.8229	16.0247	

## Data Availability

Not applicable.

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
