# Peer review of "Traumatic Impact Assessment of CPR Load on a Human Ribcage"

_ijerph, 2022, doi:10.3390/ijerph19063414_

Round 1

Reviewer 1 Report

Dear authors,

you find in attached documents all my comments and suggestions for improvement.

minor comments: in blue color

major comments: in red color

Respectfully yours.

Author Response

REPLY TO REVIEWER 1

Title: Traumatic Impact assessment Under CPR Load of a Human Ribcage

Journal: International Journal of Environmental Research and Public Health

Answer date: March 3er, 2022

Corresponding author: Christopher René Torres San Miguel

Email: ctorress@ipn.mx

We thank the reviewers for their valuable comments.

Below are the responses to the reviewers' comments regarding the comments posted.

REVIEWER #1

Q: Add supporting reference of such criteria

A: done

Q: Does this mean that the authors assumes that the load is maintained in the elastic domain of the structure

A: Yes

Q: The type and dimension of fractures has an effect. The authors should specify if there is any standard which guaranties evaluation of yield stress in case of fractured ribs

A: On the scope of this work, we are not evaluating fracture propagation. Thus, we only assess if the values reach the displacement value as an indirect measure of the probability of damage on human beings. Reference was added

Q: Be careful some of the load conditions do not seem static Corrections are need

A: You were right. We erase reference 5 and reference 13

Q: There is a shadow zone about:

  • The type of post processing operations conducted on tomography data
  • Based on figure 3, it seems that the CAD sweep is assuming a circular section of the rib. What justifies such simplification
  • What are the dimensional error between the computed tomography based geometry and the CAD sweep

A: the answers are

  • We do not preprocess the medical images
  • Reference was added, and an explanation of those considerations was also added
  • Several references mention that this parameter is variable in the human body. In order to decrease such variation, we use the following parameters: Major axis of the ellipse 8.11 mm, voxel size 0.97mm. Those values represent an image error of 7% against the reconstructed cross-section

Q: There is a non-clear zone in the modelling part. Which of the models was used in simulation, the one based on CAD Sweep or the one based on tomographic?

A: The model used in the second part of the article was the CAD Sweep model. We believe that after validating the master model of the rib (aka. The model obtained by the equations 1 to 7) we can evaluate different parameters of the rib in order to reproduce a close natural biomechanical response of the rib through artificial elements manufactured by steel, plastic or other materials. Such parameters are the shape of the cross-section, geometrical curvature, and parametric variables of position.

Q: Reviewer: What are the mechanical properties in case of orthotropic assumption

A: We add the corrections to the article

Q: Reviewer: title is false, Left image (a) What does this color field represents? Right image (b) The authors illustrate the deformed and undeformed state of which rib model. That does the color field represents. The indication on the color scale is not clear

A: You were right. The title was wrong; we already corrected it. This color field represents the stress distribution

Q: Reviewer

  • Continuous or interpolation segments have no meaning in the graph
  • Where were these displacement values extracted on which control point or zone
  • Displacement is time dependent. At which moments was each point taken to simplify better to consider the final displacements
  • The meaning or interest behind this curve is not clear at all

A: We change this Figure to increase the text's readability. The objective of the original Figure was to show the linear dependence of the payload applied versus the computed displacement, but as you suggest in the next point, we compute the stress value and also compute the strain of the model in order to compare the different set of results

Q: Reviewer. It seems that all simulations were elastic. This is what can be seem from the constant stiffnesses to see if there is a significance of table three, the authors are invited to convert the payload into stress value and to compare it to the yield stress

A: Done. Thank you for the recommendation

Q: What do the points represent? Figure is incomplete and the meaning of “Xtimes” on abscess axes is not clear. How to explain the evolution of the rib stiffness

A: We erase this Figure and add Figure 8b to increase the readability. The purpose of the original Figure was to show the stiffness interpolation parameter of modeling for the master of the rib.

Q: What do the authors mean by “damage”? It is not clear that the simulations included damage estimation or any damage criteria

Several criteria represent the damage of the thorax. One of them is the compression criteria that suggest that fractures over 52 mm of chest deflection can cause considerable damage to the bony structures of the thorax laceration of the international chest organs. If you check all the simulations, such value of displacement is not reached, so the criteria of damage can be inferred

Reviewer 2 Report

The article is interesting and valuable. The article has a correct IMRaD structure. The number of references is proper and refences are modern but you should correct references section formatting. The work is thematically consistent. The work is scientific. The presented arguments are logical. The reviewer did not notice any errors in reasoning. Introduction state the purpose of the paper. The article uses the modern finite element method.The DICOM system was used to create the CAD model. Correct mathematical inference was applied. Conclusions are correct. The reviewer enthusiastically welcomed the work in this area.

Major:

Authors should clearly articulate the research gap. Authors should refer to the achievements of other authors in the area of the analyzed issue in a more critical way.

Minor:

Edit the entire article carefully.

Put some text below equation (7) and Figure (9).

You should enlarge Figure 8.

You should increase the resolution of the drawings

Author Response

REPLY TO REVIEWER 2

Title: Traumatic Impact assessment Under CPR Load of a Human Ribcage

Journal: International Journal of Environmental Research and Public Health

Answer date: March 3er, 2022

Corresponding author: Christopher René Torres San Miguel

Email: ctorress@ipn.mx

We thank the reviewers for their valuable comments.

Below are the responses to the reviewers' comments regarding the comments posted.

REVIEWER #2

Major:

Q: Authors should clearly articulate the research gap. Authors should refer to the achievements of other authors in the area of the analyzed issue in a more critical way.

A: Thank you for the recommendation. References were added and explained in detail, this work evaluated the energy absorbed as a function of rib compression by DICOM Files and 3D reconstruction. All changes have been highlighted.

Minor:

Q: Edit the entire article carefully.

A: Done.

Q:Put some text below equation (7) and Figure (9).

A: Done.

Q: You should enlarge Figure 8.

A: Done. Figure 8 has changed

Q: You should increase the resolution of the drawings

A: Done.
